# COVID-19 Impact on Public Dental Healthcare in Bosnia and Herzegovina: Current Situation and Ongoing Perspectives

**DOI:** 10.3390/ijerph191811816

**Published:** 2022-09-19

**Authors:** Elmedin Bajrić, Amila Zukanović, Nina Marković, Amra Arslanagić, Amina Huseinbegović, Mediha Selimović-Dragaš, Sedin Kobašlija, Aleksandra Popovac, Dejan Marković

**Affiliations:** 1Faculty of Dentistry with Clinics, University of Sarajevo, Bolnička 4a, 71000 Sarajevo, Bosnia and Herzegovina; 2Faculty of Dental Medicine, University of Belgrade, Rankeova 4, 11000 Belgrade, Serbia

**Keywords:** COVID-19, Bosnia and Herzegovina, dentistry, vaccines, economic impact

## Abstract

Background: As the pandemic time went by in Bosnia and Herzegovina (B&H), various patterns toward COVID-19 itself and its impacts, implementation of prescribed preventive measures among the team members, and those of their patients, including immunization process, have been revealed. These patterns were of both empirical and evidence-based kind and consequently formed dental personnel behavior. The aim was to evaluate and compare the COVID-19 status of dentists in B&H, implementation of prescribed preventive measures, and usage of various kinds of PPE, at the beginning of the pandemics and now, 2.5 years later, including dentists’ current vaccination status, and their opinions and attitudes toward the national COVID-19 economic impact on dental practice. Methodology: Study research was in a form of a cross-sectional longitudinally designed online survey and was conducted in two parts. Results and conclusions: Dental professionals in B&H had a high frequency of COVID-19 symptoms in the second pandemic year. The vaccination status of dentists in B&H was in line with the global average values of vaccinated professionals. Dentists used patient management preventive measures and PPE recommended by WHO, but some preventive measures have been changed and prioritized recently. The economic impact of pandemics on dentistry was predominantly negative.

## 1. Introduction

After two and a half years from the COVID-19 outbreak as the global pandemic in March 2020, there are more than 541 million confirmed COVID-19 cases worldwide, with the spreading of SARS-CoV-2 in 237 countries and territories [1,2]. The dental healthcare system has not been the primary target of COVID-19 patients at the beginning of pandemics. However, dental healthcare personnel was soon pronounced to be at high risk due to the oral healthcare of patients, which was directly related to the respiratory spreading of disease. The dental practice soon has also become among the first ones to treat only those patients with emergencies worldwide, mostly in the public sector only. Lack of personal protective equipment (PPE) and disinfectant agents in the first several pandemic months also had an impact on dental practice, although the number of patients was initially significantly decreased [3,4,5,6,7,8,9,10,11].

In Bosnia and Herzegovina (B&H), national COVID-19 pandemics were declared only a few days after its global announcement, similar to neighboring and other European countries, including the strict lockdown. During this period national dental healthcare system showed decreasing visits of patients with dental emergency visits in the public sector only, and the initial lack of preventive COVID-19 resources, as well as dental materials and equipment. At the end of spring 2020, lockdown form was abandoned for good, and the functioning of the dental healthcare system started to continuously return to its basic settings in the next few months. COVID-19 positive dental healthcare personnel rapidly increased, especially until the end of June 2021. The PPE and disinfectant agents become available, but their price also rapidly continuously increased, together with the dental materials and equipment. Although dental healthcare personnel was pronounced as COVID-19 risk profession, prescribed precautionary measures for stopping of disease from spreading were mostly of no difference compared to the pre-COVID-19 period considering their usage and/or implementation in B&H. Truly, some of the prescribed PPE forms (different kinds of gowns and FFP2/FFP3 masks, for example) were previously, in the pre-COVID-19 period, used only in some of the hospital intensive care units and/or clean clinical laboratory conditions. As the pandemic time went by, various patterns toward COVID-19 itself and its impacts, implementation of prescribed preventive measures among the team members, and those of their patients, have been revealed. These patterns were of both empirical and evidence-based kind and consequently formed dental personnel behavior [1,2,12,13,14,15,16,17].

The passive immunization process against SARS-CoV-2 in B&H has begun during the winter of 2021, with dental healthcare personnel among the first ones to be immunized. Up until now our country confirmed almost 386 000 COVID-19 positive cases, with over 1.92 million vaccine doses administered, and a national vaccination rate of almost 26% of the population only. COVID-19 immunization process was established only on a voluntary basis mostly worldwide, including in our country. Furthermore, general national public opinion toward the immunization process has already in the pre-COVID-19 era been compromised and mostly influenced by non-medical and/or quasi-medical opinions and individuals, where even some of the healthcare workers have taken their participation. This has resulted not only in temporary re-emergence of already eradicated child infectious diseases, but also in consequently increased child lethality [12,13,14,18].

Based on previous studies, the aim of this study was to evaluate and compare the COVID-19 status of dentists in B&H, implementation of prescribed preventive measures, and usage of various kinds of PPE, at the beginning of the pandemic and now, 2.5 years later. Moreover, the aim was to determine and evaluate dentists’ current vaccination status, as well as their opinions and attitudes toward the national COVID-19 economic impact on dental practice.

## 2. Materials and Methods

The study research was self-supported and in a form of a cross-sectional longitudinally designed online survey and was conducted in two parts. The first part of the survey was performed within the COVIDental Collaboration Group in B&H as one of the 35 countries involved, with the study survey protocol previously described in detail [19]. In brief, an anonymous online survey using Google forms was sent electronically to dentists of our country, which was composed of 25 questions in total, mostly regarding their demographic characteristics, COVID-19 status (level and frequency of acquired symptoms, knowledge and attitudes toward SARS-CoV-2 infection), and level of performing prescribed protective measures and usage of PPE, at the beginning of the pandemics and in perspective through near future. National dental chambers supported this research by sending e-mails with links to a survey in the following way: the Dental Chamber of the Federation of Bosnia and Herzegovina sent e-mails to 1090 registered dentists and the Dental Chamber of the Republic of Srpska sent e-mails to 1143 registered dentists. The survey itself has been conducted within 15 days period, from 29 June until 14 July 2020.

The second part of the study has been conducted recently, with the same study protocol as the previous first part from 2020. The survey form was reduced and also enhanced at the same time, and finally composed of 20 questions in total. Besides COVID-19 status and level and usage of prescribed measures and PPE of surveyed dentists, questions regarding their vaccination status and attitudes about COVID-19 economic impact were also added. The online survey lasted this time for 39 days, from 11 May until 19 June 2022.

The study was conducted in accordance with the Declaration of Helsinki. Ethical approval for these kinds of anonymous questionnaires was not required by law in our country. Surveyed dentists of both study parts electronically signed informed consent at the beginning of each online survey form as a mandatory option.

Statistical analyses of obtained results were performed in the following way:-Demographic characteristics of the study sample, results regarding participants’ COVID-19 status, level and usage of prescribed measures and PPE, vaccination status, and attitudes about the economical impact of COVID-19, were analyzed by their total and relative counts and are presented in tables and charts;-Significant differences between obtained study results were determined using the Chi-square test.

All statistical analyses were performed with the Microsoft Excel software v. 2019 and IBM Statistical Package (IBM, Armonk, NY, USA) for Social Sciences software v. 23 (SPSS Inc., Chicago, IL, USA) for the Windows operative system, with the predetermined significance set to the value of *p* ≤ 0.05.

## 3. Results

The study sample included 349 dentists in total. The first sample part included back-sent forms from participants surveyed in 2020. The second sample part was composed of back-sent survey forms from 136 participants surveyed in 2022. In Table 1, there are several descriptive and demographic characteristics of the study participants.

Accordingly, the sample covered the whole country, where study participants were mostly general female dentists from private practice.

The 2020 study sample was surveyed almost at the beginning of the COVID-19 pandemics in our country, with more reported COVID-19 referable symptoms than COVID-19 confirmed cases. The COVID-19 prevalence in the study sample was obviously increased two years later in COVID-19 confirmed participants only, where more than a third of study participants did not get sick so far (Table 2).

The distribution of most usual symptoms of COVID-19 in the 2020 and 2022 total samples is presented in Figure 1.

It has already been expected that the prevalence of most usual COVID-19 symptoms significantly increased after two years, which was confirmed and is presented in Table 3.

Although the immunization process recently offered the most efficient tool against further spreading of the disease and expression of its hardest clinical forms, there was still a certain serious number of dentists who refused to be vaccinated (Table 4). Dentists mostly expressed their disbelief in the immunization process and efficacy of a COVID-19 vaccine.

At the beginning of the pandemic, the COVID-19 management measures for dental patients were considered a crucial means of stopping the disease from spreading within the dental practice. Dentists mostly considered that these preventive measures had to be implemented in a more intensive manner in the (near) future, even more than they did at the beginning of the pandemic. After the implementation of the survey in 2022, it was clearer that 2020 perspectives were set mostly higher and that some of them were even abandoned nowadays. Nevertheless, the implementation of these measures in the 2022 sample was still higher than at the beginning of the pandemic (Figure 2). In the 2022 study sample, there was a significant number of surveyed participants (41.90%) who also have taken part in the survey in 2020. However, their answers have not been statistically different at all regarding their current attitudes about the implementation of prescribed preventive measures compared to other participants from the 2022 sample. However, statistically significant differences which were determined within the 2020 sample and between the 2020 and 2022 samples are shown in Table 5.

Professional protection equipment (PPE) was considered the main personal tool for stopping the SARS-CoV-2 infection in staff members at dental workplaces. Standard and new PPE forms were introduced to dental practice at the beginning of the pandemic (Figure 3). Similarly, implementation rates of various PPE forms mostly decreased, not only in future perspective point of view but also even more in the 2022 sample (Figure 3). Statistically significant differences which were determined within 2020 and between 2020 and 2022 samples are shown in Table 6. Similar to the implementation rates of prescribed measures for COVID-19 prevention, statistically significant differences have not been determined within the 2022 study sample participants regarding the usage of PPE.

Empirical and evidence-based knowledge about the SARS-CoV-2 infection and COVID-19, which was formed in the last 2.5 years of the pandemic shaped the opinion of the study participants within this period (Table 7). Awareness of the risk for the dental profession which could not be easily avoided still existed and was lower than at the beginning of the pandemic (Mann–Whitney U = 10,144.00, *p* ≤ 0.0005).

Figure 4 showed that the economic influences of the COVID-19 pandemic inevitably targeted also dental practice and patients, in the way of significantly increasing costs of dental materials and equipment and dental services as well. Subsequent lack of assets in dental patients additionally caused a significant decrease in dental visits.

## 4. Discussion

This was the first comprehensive study in the Balkan countries about the impact of the COVID-19 pandemic on dental practice, as well as the change of perspective from the beginning of the pandemic and now, 2.5 years later. Dental professionals in B&H had a high frequency of COVID-19 symptoms, mostly during the second pandemic year. They used patient management preventive measures and PPE recommended by WHO, but some preventive measures have been changed and prioritized recently. The economic impact of the pandemic on dentistry was predominantly negative.

The frequency of dental professionals in B&H who had symptoms of COVID-19 was quite high during the second pandemic year (64.7%). This was understandable because, after the lockdown, dentists have worked regularly. This may be an indicator that preventive measures and PPE did not provide sufficient protection against SARS- CoV-2, but on the other hand, it was likely that dentists got COVID-19 through private contacts, outside of working places. Given that there was no contact tracing trend in B&H after the first few pandemic months, it was almost impossible to find out how dentists became infected. Data from a global study showed about 15% of dentists had COVID-19 infection during the first pandemic year [20]. This could correspond to our findings, since there were 13.6% of dentists in B&H with COVID-19-like symptoms during 2020, although only 0.47% cases were confirmed. This discrepancy could be explained with lack of COVID-19 PCR and antigens tests during the first pandemic months in B&H, which resulted in some unconfirmed cases. Moreover, as discussed further, at the beginning of the pandemic, more dentists experienced anxiety which could cause somatization and focus on cold and flu symptoms. The large increase of infected people after 2020 showed that in spite of the application of protective measures and vaccination programs for the population, new emerging variants of the virus could mean that countries would be still at risk.

The most frequent symptoms at the beginning (2020 sample) were headaches and fatigue, followed by upper respiratory symptoms, which were similar to the findings of others [21,22]. The symptoms in 2022 sample have significantly risen, but they also have qualitatively changed in a general and neuromuscular form, which correlated with the SARS-CoV-2 variants dominant in B&H in 2020 (mostly alpha and beta variants) and in 2022 (mostly lambda and omicron variants) [2,12,13,14].

Contrary to the fact that dentists were infected much less often in the first pandemic year, their assessment of high professional risk was 64.8% in the first year, while in the second year it was moderately lower, 38.2%. The lower risk assessment can be explained by the greater accessibility of information about COVID-19 infection, the availability of the vaccine, and less working anxiety in the second pandemic year compared to the beginning of the pandemic [23,24]. On the other hand, the assessment of the possibility to protect oneself from COVID-19 infection was approximately the same at the beginning of the pandemic as recently (pretty unsure 55.4% in 2022 and 59.6% in 2022). This was reality, considering new virus strains and mutations which enable its break through PPE and vaccination immunity. Similar were the results of a study that showed that only 36.8% of dentists felt confident about being able to prevent infection transmission in the dental setting [25].

The vaccination status of dentists in B&H was in line with the global average value with 86% of vaccinated professionals. Similar values of acceptance of the COVID-19 vaccine among dentists were obtained by Gopakumar et al. (85%) [26], and 81% in the global study [27], with the highest percentage in Kuwait (91%) and the lowest in Pakistan (50%). Although the representation of vaccinated dentists corresponded to the global average, the existence of about 5% of them who did not believe in vaccines, and another 3% who did not consider that there were reasons for COVID-19 vaccination, may be of serious concern. Furthermore, few studies showed much higher hesitation in COVID-19 vaccine acceptance among dental students compared to graduated dental professionals. In a global study, the COVID-19 vaccine acceptance rate was found to be 60% and in another 22.5% of dental students worldwide were hesitant, and 13.9% rejected COVID-19 vaccination [26,27,28,29]. These results have indicated that it should be necessary to increase the scientific awareness of the student population in the future.

Preventive measures in dental practice have always been applied, but with the beginning of the COVID-19 pandemic, they gained much more importance, due to the fact that the COVID-19 vaccine was not expected to be implemented in the further period from the 2020 study sample point of view. That could also be seen in our results from 2020 when dentists applied numerous preventive measures, mostly telephone triage (73.5%), handle disinfection (63.9%), hand disinfection of patients (60.2%), ventilation of the waiting room (60.2%), and hand disinfection of dentist (64.5%). This was mostly in accordance with the results from a global study [20] where the most common patient preventive measures were found to be “reduce crowding in the waiting room” (up to 93.8% compared to 75.5% in our study) and the “patients’ health status and body temperature”, as checked by 57.6% of the respondents (48.2% in our study). These measures were slightly more applied in other countries compared to B&H, especially in countries with higher gross national income (Germany and USA) [20]. In a Belgian study, the prevention of cross-contamination was done in 74% by checking the patient’s general health status together with rinsing the oral cavity with 1% H_2_O_2_ or 1% iodine polyvidone, and the least frequent measures were physical distance in the waiting room (55.8%) and checking body temperature (36.7%) [25]. This was not totally in accordance with our and global results and showed the diversity and inconsistency of the applied preventive measures.

While some preventive measures were not new, such as disinfection of surfaces in the dental office, ventilation, and disinfection of the doctor’s hands, many measures were specifically introduced as protection against COVID-19 and were not common in dental offices in the pre-COVID-19 period. This primarily referred to physical distance in the waiting room, a patient wearing masks in the waiting room, waiting outside the dental office, not using aerosol-producing handpieces, and rinsing the mouth with antiseptics. The use of mouth rinses intended to reduce SARS- CoV-2 salivary load, and generally it was highly variable, from alcohol, and povidone-iodine to essential oils or cetylpyridinium chloride-containing mouth rinses. The differences in pre-COVID-19 and COVID-19 preventive measures indicated the importance of assessing the participants’ opinions about the future perspectives for the conduction of these measures. Based on knowledge gained during the first months of the pandemic, dentists believed that some measures such as “disposal of single-use and disinfection of multiple-use equipment”, “hand disinfection of dentist”, “surface disinfection with alcohol”, and “time and space in the waiting room” would be needed more, but without statistically significant difference. Other measurements were assessed to be significantly less important such as “phone triage”, “postponement of the treatment”, and “not using aerosol producing handpieces”. These last assessments proved to be accurate and logical, given that in the first pandemic months only emergency dental interventions were performed, so with the re-establishment of the normal work regime, it was logical to reject or reduce these measures. On the other hand, the assessment for many measures to become more important was overestimated. The difference between 2020 and 2022 results showed significantly less usage of mouth rinses with H_2_O_2_ and povidone-iodine, surface disinfection with active ingredients solutions, hand disinfection of dentists, disposal of single and disinfection of multiple-use equipment, and not using of aerosol producing handpieces. Although more recent knowledge about the spread of the SARS-CoV-2 virus as well as mass vaccination caused a reduced and different application of preventive measures compared to the beginning of the pandemic, it was obvious that preventive measures would remain at a higher level compared to the period before the pandemic.

During the first pandemic months, dentists in B&H most frequently used visors, non-surgical masks, and non-sterile gloves for PPE. Although the use of N95 masks was emphasized as the main recommendation for PPE, it was known that there was a lack of these resources in 2020 worldwide. Moreover, it was possible that there has been a certain degree of aversion to the “new” PPE (N95, gowns, goggles), especially among elderly doctors. In a Belgian study, dentists aged 55 years or above implemented guidelines related to patient protective management and PPE significantly less frequently [25]. Results from the global study showed more frequent use of N95 masks in dentists (53% compared to 42.2% in our study). Furthermore, reported routine use of FFP2/N95 masks was the only measure significantly associated with a reduced risk of acquisition of viral infection by dentists [20]. Belgian dentists reported wearing FFP2/N95 masks in 82.2%, while surgical masks only in 16.9% of cases. Comparison between 2020 and 2022 found significantly reduced usage of visors, gowns, caps, and “prolonged use of PPE”. Similar to protective measures, reduced use of PPE could be explained by mass vaccination, less anxiety in dental professionals, and also by a large number of previous COVID-19 infections among dentists which made them feel “protected”. Carvalho and associates concluded that dentists having experienced COVID-19 reported a high self-perceived risk of virus acquisition, a lower concern of getting infected, and lower confidence in being able to prevent disease transmission in the dental setting [25].

The negative economic impact of the pandemic on dental practice was visible and present after 2.5 years from its beginning, through a significant increase in the cost of expenses, and on the other hand, working with the same or fewer number of patients at the same or slightly increased service prices. In a study assessing the dentists’ attitudes toward the impact of the pandemic on dental practice in England, participants from public sector expressed concern about difficult patient access and the subsequent backlog of emergency cases, while those in private sector expressed concerns about practice sustainability [30,31].

Limitation of study results was caused primarily by the low availability of recent global data on the current COVID-19 situation in dental practice, i.e., how priorities in preventive measures, immunization, and economic impact could change as the pandemic progresses. Another important study limitation was the online surveying itself. These self-evaluation methods were usually offering a fast and convenient way to conduct a study, and no matter how well methodologically designed, there was always a potential risk for bias in sample size, responding, and collecting the results as well [32]. Normative values of our online survey form have only been partially determined, with test-retest reliability following the original study, but without face- and content validation which should be improved in future studies [19,33]. Relatively low response rates in our samples could be generally considered as obstacle, but they were still regarded reliable in online surveying forms [19,34]. This response dropout could be also related to possible ethical wavering of our participants, which was generally present in COVID-19 online surveying [35].

COVID-19 has changed dental practice worldwide in terms of priorities, job risks, and also business methods, and perspectives. It became obvious that there was no way for a quick and effective way to eradicate these and similar viruses. So, the functioning of dental practices in both the public and private sectors needs to adapt. It should be necessary that relevant institutions and associations, dental chambers, and global stakeholders become more actively involved in developing a strategy to provide a support framework, in order to achieve benefits for patients, reduce risks for dentists, and help maintain business.

## 5. Conclusions

Dental professionals in B&H had a high frequency of COVID-19 symptoms in the second pandemic year. The vaccination status of dentists in B&H was in line with the global average values of vaccinated professionals. Dentists used patient management preventive measures and PPE recommended by WHO, but some preventive measures have been changed and prioritized recently. The economic impact of the pandemic on dentistry was predominantly negative.

## Figures and Tables

**Figure 1 ijerph-19-11816-f001:**
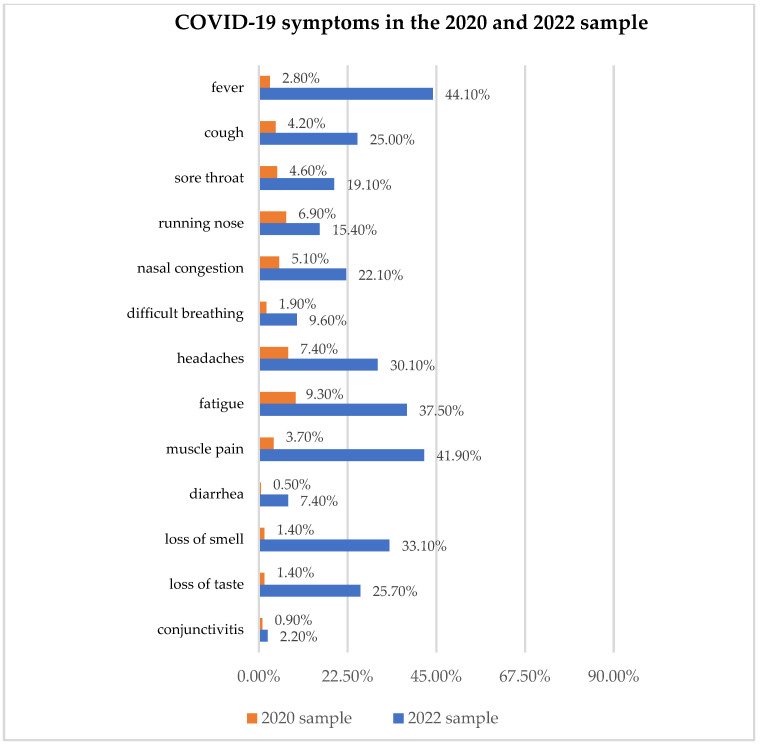
Presence of COVID-19 symptoms in study participants.

**Figure 2 ijerph-19-11816-f002:**
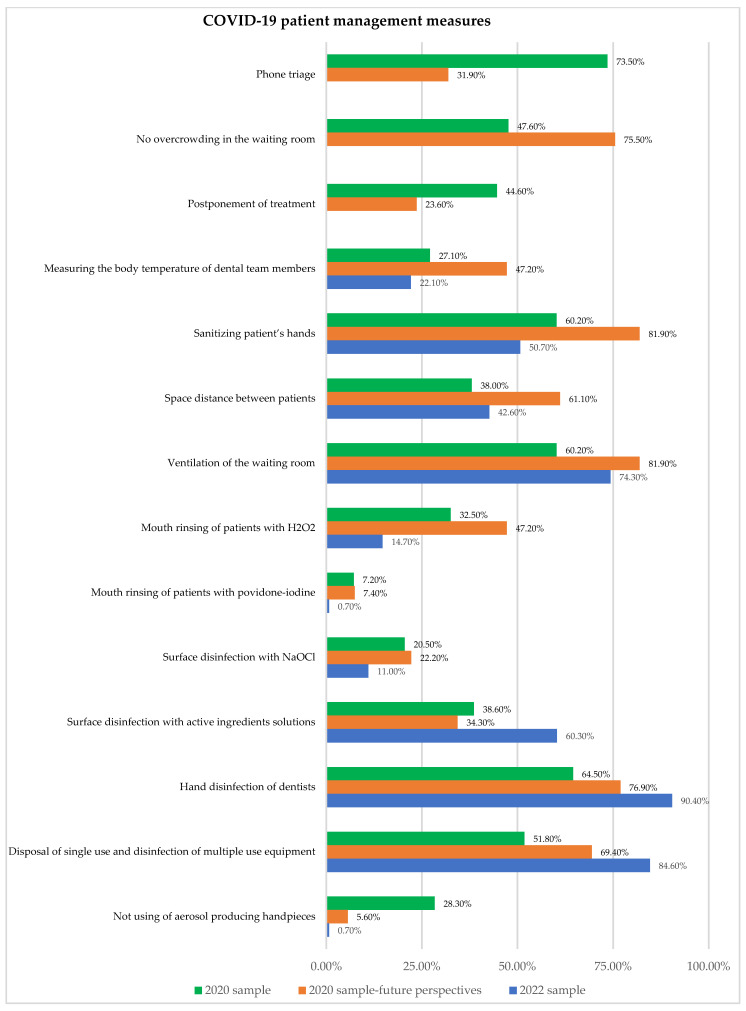
Presence of COVID-19 patient management measures in study participants which mostly differed.

**Figure 3 ijerph-19-11816-f003:**
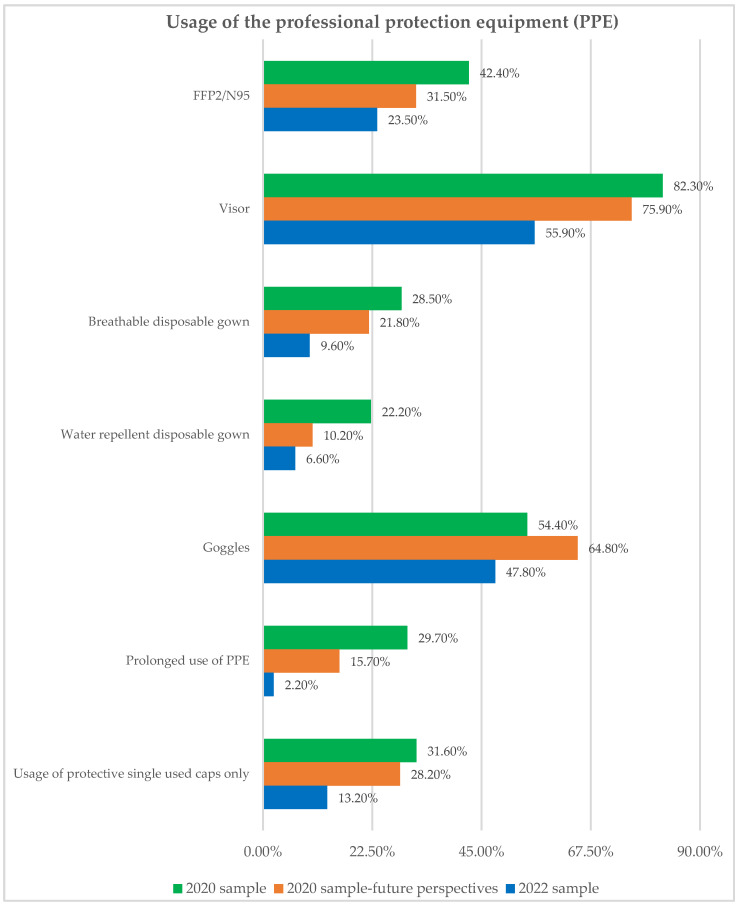
Usage of various PPE forms in study participants which mostly differed.

**Figure 4 ijerph-19-11816-f004:**
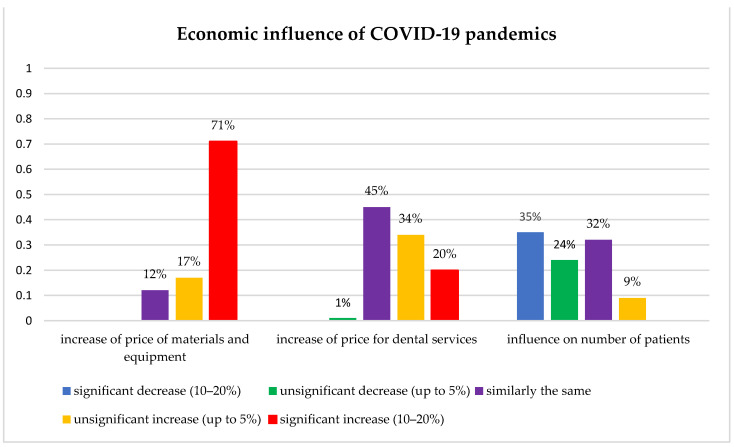
Economic influence of COVID-19 pandemics on dentistry in B&H.

**Table 1 ijerph-19-11816-t001:** Descriptive and demographic characteristics of the study sample.

**Descriptive Characteristics**				
**Number of** **Surveyed Dentists**	**2020 Sample**		**2022 Sample**	
	**Frequency**	**%**	**Frequency**	**%**
Number of surveyed dentists	2233		2233	
Number of fulfilled forms	213	9.54	136	6.09
Total	213	9.54	136	6.09
**Gender of study participants**	**2020 sample**		**2022 sample**	
	**Frequency**	**%**	**Frequency**	**%**
Male	64	30.00	31	22.80
Female	149	70.00	105	77.20
Total	213	100.00	136	100.00
**Age of** **study participants**	**2020 sample**		**2022 sample**	
	Mean	Std. Deviation	Mean	Std. Deviation
	38.2911	8.07043	41.0735	8.16372
**Demographic characteristics**				
**Workplace of** **study participants**	**2020 sample**		**2022 sample**	
	**Frequency**	**%**	**Frequency**	**%**
Federation of B&H	126	59.20	82	60.30
Republic of Srpska	84	39.40	54	39.70
Brčko district	3	1.40	―	―
Total	213	100.00	136	100.00
**Working status of** **study participants**	**2020 sample**		**2022 sample**	
	**Frequency**	**%**	**Frequency**	**%**
Private practice (owner)	58	27.20	44	32.40
Private practice (salaried)	65	30.50	38	27.90
Private and public sector	11	5.20	4	2.90
Public sector	73	34.30	42	30.90
Academic/research	6	2.80	8	5.90
Total	213	100.00	136	100.00
**Dentistry field of** **study participants**	**2020 sample**		**2022 sample**	
	**Frequency**	**%**	**Frequency**	**%**
General dentist	139	65.30	77	56.60
Dental specialist	74	34.70	59	43.40
Total	213	100.00	136	100.00

**Table 2 ijerph-19-11816-t002:** COVID-19 presence in study participants.

**Confirmed COVID-19 Cases**			
**2020 Sample**		**2022 Sample**	
**Frequency**	**%**	**Frequency**	**%**
1	0.47	88	64.70
**Report of suffering COVID-19 referable symptoms**			
**2020 sample**		**2022 sample**	
**Frequency**	**%**	**Frequency**	**%**
29	13.61	88	64.70

**Table 3 ijerph-19-11816-t003:** Differences in COVID-19 symptoms between the 2020 and 2022 sample.

COVID-19 Symptom	Chi-Square	*p*	COVID-19 Symptom	Chi- Square	*p*
Fever	92.327	*p* ≤ 0.0005	Headaches	31.121	*p* ≤ 0.0005
Cough	33.161	*p* ≤ 0.0005	Fatigue	40.474	*p* ≤ 0.0005
Sore throat	18.664	*p* ≤ 0.0005	Muscle pain	79.733	*p* ≤ 0.0005
Running nose	5.403	*p* = 0.020	Diarrhea	12.884	*p* ≤ 0.0005
Nasal congestion	22.851	*p* ≤ 0.0005	Loss of smell	70.226	*p* ≤ 0.0005
Difficult breathing	10.568	*p* = 0.001	Loss of taste	50.626	*p* ≤ 0.0005

**Table 4 ijerph-19-11816-t004:** COVID 19 vaccination status in the 2022 sample.

**Status**	**Frequency**	**%**	**Dose**	**Frequency**	**%**
Not vaccinated	19	14.00	1st dose	2	1.50
Vaccinated	117	86.00	2nd dose	59	43.40
Total	136	100.00	3rd dose	56	41.20
**Reasons not to take the COVID-19 vaccine**				**Frequency**	**%**
Personal medical reasons				8	5.90
Disbelief in COVID-19 vaccine efficacy				6	4.40
General disbelief in vaccination				1	0.70
No reasons for COVID-19 vaccination				4	2.90

**Table 5 ijerph-19-11816-t005:** Differences in COVID-19 patient management measures within the 2020 sample and between the 2020 and 2022 samples.

Differences within the 2020 Sample			Differences between the 2020 and 2022 Sample		
Measure	Chi-Square	*p*	Measure	Chi-Square	*p*
Phone triage	86.014	*p* ≤ 0.0005	Ventilation of the waiting room	7.008	*p* = 0.008
No overcrowding in the waiting room	11.093	*p* = 0.001	Mouth rinsing of patients with H_2_O_2_	13.283	*p* ≤ 0.0005
Postponement of treatment	29.259	*p* ≤ 0.0005	Mouth rinsing of patients with povidone-iodine	7.769	*p* = 0.005
Measuring the body temperature of dental team members	6.568	*p* = 0.010	Surface disinfection with NaOCl	5.121	*p* = 0.024
Sanitizing patient’s hands	4.264	*p* = 0.039	Surface disinfection with active ingredients solutions	15.902	*p* ≤ 0.0005
Space distance between patients	8.835	*p* = 0.003	Hand disinfection of dentists	27.409	*p* ≤ 0.0005
Not using of aerosol producing handpieces	44.558	*p* ≤ 0.0005	Disposal of single and disinfection of multiple-use equipment	37.004	*p* ≤ 0.0005
			Not using of aerosol producing handpieces	41.980	*p* ≤ 0.0005

**Table 6 ijerph-19-11816-t006:** Differences in the usage of PPE within the 2020 sample and between the 2020 and 2022 samples.

Differences within the 2020 Sample			Differences between 2020 and 2022 Sample		
PPE	Chi-Square	*p*	PPE	Chi-Square	*p*
FFP2/N95 mask	4.197	*p* = 0.040	FFP2/N95 mask	11.490	*p* = 0.001
Water repellent disposable gown	9.197	*p* = 0.002	Visor	23.632	*p* ≤ 0.0005
Goggles	3.996	*p* = 0.046	Breathable disposable gown	15.197	*p* ≤ 0.0005
Prolonged use of PPE	10.547	*p* = 0.001	Water repellent disposable gown	13.328	*p* ≤ 0.0005
			Prolonged use of PPE	39.918	*p* ≤ 0.0005
			Protective caps single use	14.399	*p* ≤ 0.0005

**Table 7 ijerph-19-11816-t007:** Attitudes of the study participants toward the SARS-CoV-2 infection.

**Professional Risk of Infection for Dentists**	**2020 Sample**		**2022 Sample**	
	**Frequency**	**%**	**Frequency**	**%**
Very unlikely	4	1.90	6	4.40
Unlikely	7	3.30	20	14.70
Likely	64	30.00	58	42.60
Very likely	138	64.80	52	38.20
Total	213	100.00	136	100.0
**Possibility of avoiding infection**	**2020 sample**		**2022 sample**	
	**Frequency**	**%**	**Frequency**	**%**
Totally sure	5	2.30	3	2.20
Pretty sure	25	11.70	15	11.00
Pretty unsure	118	55.40	81	59.60
Totally unsure	65	30.50	37	27.20
Total	213	100.00	136	100.0

## Data Availability

Not applicable.

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
