# Peer review of "COVID-19 Impact on Public Dental Healthcare in Bosnia and Herzegovina: Current Situation and Ongoing Perspectives"

_ijerph, 2022, doi:10.3390/ijerph191811816_

Round 1

Reviewer 1 Report

This is a complete study, with interesting information
I will suggewt to remove figure 10, and correct the other tables to start the words with capittal letrers

This research assesses the COVID pandemic situation in dentistry in Balcanic countries through surveys in two different pandemic moments, during the lockdown and two years after.

I consider the topic is relevant in the field, maybe not original as there are other similar works in other countries, because it is interesting to evaluate consequences in different scenarios. Each country has developed its own rules and clinical decisions to solve COVID situations, and registering and comparing those results are relevant to make future decisions. Usually, dentistry is developed in private clinics, and governments don't make too many efforts to assess clinical security or better methods to safely develop our daily clinical practices.

This manuscript makes 2 surveys separately for 2 years, permit to compare the evolution of the situation and also the population included is original.

The surveys are complete and address different issues, from the issue of safety in daily practice, the economic impact and individual beliefs that influence the vaccination of the population studied.

Conclusions are consistent and arguments address the main question posed.

The references are appropriate.

Reviewer 2 Report

Thank you for the manuscript. It was an interesting read, and of worth to people within the field as it may also inform decision-making processes. 

Language-editing is needed by a native English speaker to ensure that reading is streamlined, and meaning is clear.

Please define what is considered Spring 2021.

I suggest the questionnaire be made available for review, as it may influence scientific rigour.

Would it be possible to provide evidence of ethics wavering for such a study?

Was the survey face- and content-validated?

What sample size could be considered generalisable considering the large amount of dentists originally surveyed, but not completed?

Where in the data does 'respectful period' in practice show? Is this considered based on their age? This will not be a direct link.

Were participants not provided the opportunity to choose a gender outside of the binary?

What was considered mild or severe symptoms?

Given the different variants that spread between regions, is there a possible link to this given the symptomatic profiles? I recommend these be discussed.

Reviewer 3 Report

Dear Authors,

Thank you for the opportunity to review your manuscript. The paper is timely and sheds light on an ongoing issue faced by the dental service community. 

The paper does need some additional work to showcase the value of the findings. Please see below: 

1) Introduction - it is not clear from the introduction as to what the highlight of the rest of the paper is. It would help to narrow down the focus to COVID-19 issues within the dental community globally, regionally and B&H. Please also pay focus on English grammar, tense and framing as it is difficult to understand the meaning of sentences - this applies to the entire paper. 

2) Materials and methods - if possible please add the sever questions (of a sample of it) as an appendix to give a clear picture of what was asked. 

3) Results - 

- tables 1 and 2 can be combined to represent the demographic characteristics of the sample population in 2020 and 2022. 

- since only 1 respondent denoted to have had covid in the 2020 cycle, the graphical presentation of COVID-19 symptoms in figure 1 as well as the differences denoted in table 4 do not make sense. It is obvious that due to the extremely small sample of N=1 in 2020, the chi square tests are significant in table 4. It should suffice to only present symptoms for 2022 sample and remove table 4 from the paper. in chart 1, you probably want to show the top 5 symptoms instead of listing all the possible symptoms. I am also not sure what chart 2 is trying to highlight as the title of paper is on the impact within dental community while the symptoms of COVID are not unique to the dental providers. 

- It is difficult to understand table 6 and chart 3. Again, it might help to focus on the top few chosen measures instead of listing out all the responses as a way to declutter the table and chart. right now, I cannot tell clearly from the table what measures were more used than others. Same goes for chart 4 and table 7. 

- might help to combine tables 8 and 9, and to move some charts (5 - 7) to the appendix to reduce the number of tables and figures in the main manuscript. 

4) discussion - please rewrite the section to focus on the main findings and English language edit.

Good luck. 
